# Prevention of Migration of Esophageal Self-Expandable Metallic Stents Using Endoscopic Clips

**DOI:** 10.3390/medicina59112035

**Published:** 2023-11-17

**Authors:** Nikola Boyanov, Katina Shtereva, Katerina Madzharova, Liuben Kirkov, Neno Shopov, Vladimir Andonov

**Affiliations:** 1Medical Simulation Training Center Research Institute at Medical University of Plovdiv, 4000 Plovdiv, Bulgaria; nikolaboyanov@gmail.com; 2Department of Gastroenterology, Pulmed University Hospital, 4000 Plovdiv, Bulgaria; 3Department of Surgery, Pulmed University Hospital, 4000 Plovdiv, Bulgaria; 4Second Department of Internal Medicine, Medical University of Plovdiv, 4000 Plovdiv, Bulgaria; 5Department of Gastroenterology, Kaspela University Hospital, 4001 Plovdiv, Bulgaria

**Keywords:** esophageal obstruction, self-expandable metallic stent, clip fixation, migration

## Abstract

*Background and Objectives*: Esophageal stenting with self-expandable metallic stents (SEMS), for both benign and malignant reasons, has been widely practiced for decades, but migration still remains the most common complication of the procedure. In this report we aim to review our experience and results in stent fixation with clips. *Materials and Methods*: We present 18 patients who underwent esophageal stenting for both benign and malignant reasons. The SEMSs used were partially covered and were fixated with two to four through the scope hemostatic clips in the proximal end of the prothesis. The procedure was performed only on patients with a high risk of migration of the stent. *Results*: Migration occurred in only one of the above-mentioned patients and was treated with stent repositioning. The other adverse events that occurred were related to tumor growth in patients with malignant diseases. *Conclusions*: Clip fixation of an esophageal self-expandable metallic stent in cases considered high-risk for migration is a safe procedure. It reduces the migration rate significantly for both benign and malignant indications.

## 1. Introduction

Placement of a self-expandable metallic stent (SEMS) in both benign and malignant strictures of the esophagus has been widely practiced for decades [1]. According to the European Society of Gastrointestinal Endoscopy’s (ESGE) latest guideline on stenting in the upper gastrointestinal tract, SEMSs are considered first line treatment for patients with malignant obstructions and short-term life expectancy or in combination with brachytherapy for longer life expectancy. Moreover, metallic stents have a pivotal role in the management of benign strictures, perforations and fistulas of the esophagus [2]. Stent migration remains the most common complication of the procedure. Benign reasons for stenting, including perforations, fistulas and strictures, have a higher risk of migration (30–62%) [1,3]. Patients with malignant disease at high risk of migration are those who are undergoing brachytherapy, as well as patients with distal strictures close to the gastroesophageal junction [1,4].

In this study, we review our experience and results in esophageal stenting and using through-the-scope hemostatic clips.

## 2. Materials and Methods

In the period from July 2018 to July 2023, in our tertiary medical center in Plovdiv, Bulgaria, we placed over 70 SEMSs in cases with both benign and malignant conditions. Due to the complication of migration, in the year of 2020, our team started fixating the proximal edge of the stent with through-the-scope hemostatic clips in the cases with higher risk for migration.

We present a total of 18 patients—12 male and 6 female, aged 41 to 84, that have undergone the procedure of SEMS placement with clips fixation, due to different reasons, shown in Table 1. Overall, out of the 18 cases, 13 had malignant obstruction due to adenocarcinoma of the cardia and/or distal esophagus [7], esophageal squamous cell carcinoma [1] and obstruction caused by mediastinal lymph nodes or progression of lung cancer [3]. The other five patients had benign reasons for stenting: perforations [2], benign strictures [2] and one broncho-esophageal fistula. Written consent was obtained from all patients prior to the procedure. All of the patients with malignant disease have been previously diagnosed and histologically verified.

Our team used Partially Covered Esophageal Stents (Changzhou City Zhiye Medical Devices Institute, Changzhou, China and Wallflex, Boston Scientific, Boston, MA, USA) (Figure 1), which are both distal opening stents. A scarification test for Urografin sensibility took place prior to the manipulation. The procedure was performed under general anesthesia. The location and length of the stenosis was determined with endoscopy prior to the stenting. A submucosal injection of a solution of Urografin was placed 1 to 2 cm above the proximal end of the stricture and was used as a marker for the position of the SEMS. The stent was positioned over a guidewire and deployed under fluoroscopy guidance and its position and the efficiency of the procedure were verified through endoscopy. The SEMS was fixed in its proximal end using 2 to 4 EZ through-the-scope clips (Olympus) (Figure 2). The stent fixation was performed under endoscopic guidance and the clips were positioned either on two opposite sides or in the resemblance of the Mercedes sign. We used two clips in nine cases, three clips in eight cases and four clips in only one case.

The length of the stent was chosen in correlation with the length of the stenosis. The stents we used were 20 mm in diameter and either 100 mm in length (Changzhou City Zhiye Medical Devices Institute, China) or 12 mm in length (Wallflex, Boston Scientific, MA, USA). The SEMS needed to be at least four centimeters longer than the obstruction. The following day, radiography with peroral contrast was performed to verify the correct position of the stent. The number of the clips depended on the reason for stenting. Typically, malignant obstruction with a higher grade of stricture is at lower risk of migration, so in those cases we placed a lesser number of clips (2 clips). In cases with benign reasons for stenting or malignant obstruction that is not that prominent, we placed a higher number of clips.

## 3. Results

In the malignant portion of the cases, three complications were reported. Only one of them was proximal migration of the prothesis noted in a patient with cardiac cancer and esophageal obstruction two weeks after the procedure. The migration resulted in repositioning of the SEMS. The other two complications were associated with tumor growth. In one of the patients with lung cancer and esophageal obstruction, tumor growth through the SEMS was noted. In the other case, tumor growth had spread higher than the proximal end of the stent and a second SEMS was placed overlying the first one. The follow-up period for the malignant cases was three months. No other short or long-term complications associated with the SEMS or the clip fixation were reported.

In the benign section of the research, no migrations or other complications were reported. The follow-up period for those cases was one year. The metallic stents were removed in a period from two weeks to two months, depending on the reason for stenting. There were no complications associated with the removal of the SEMS.

## 4. Discussion

The complications of esophageal stenting are divided into two groups—early and late complications. The early ones include pain, perforation, bleeding and technical failure in the deployment of the stent, whilst the late ones include migration of the stent, tissue ingrowth/overgrowth and clinical complications (recurrent dysphagia, gastrointestinal reflux, etc.) [5,6]. Migration is the most common complication after stent placement, ranging from 4–38% [4,7,8]. Typically, the fully covered SEMSs are more prone to migration than the partially covered and the non-covered ones [4]. The covering of the fully covered stents prevents embedding of the stent into the esophageal wall, which leaves them more prone to migration. Different kinds of anti-migration features were developed, ranging from means of outer fixation to shouldered ends, double-layer design, and endoscopic suturing and fixation [9]. In a study including 61 patients with both benign and malignant diseases, esophageal stents were externally fixated using a silk tread. No migrations occurred in this study, but external fixation adds additional discomfort for the patients [10]. A systematic review of six studies, consisting of 250 patients with malignant obstruction treated with double-layered SEMSs, showed 4.7% of migrations [11]. Endoscopic suturing and over-the-scope clip rates of migration were compared to those of non-fixated fully covered SEMSs. The results of this study show 32% risk of migration with endoscopic suturing and 12% risk with the over-the-scope clips, compared to 55% in the cases without fixation [12].

In the occasion that the stent is placed distally of the gastroesophageal junction or there is a lack of a stenosis in the esophagus (perforation, fistula), there is a higher risk of migration of the SEMS [13]. Even though methods of stent fixation significantly reduce the incidence of migration in these situations, a comparative study of 44 patients suggests that there is still a 13% risk of migration of the prothesis after clip fixation [14]. In our study, only one of eighteen procedures resulted in stent migration (5.5%), which shows even better results than the above-mentioned study [14].

## 5. Conclusions

Clip fixation of an esophageal self-expandable metallic stent in cases considered high-risk for migration is a safe procedure. It reduces the migration rate significantly for both benign and malignant indications. Further research with a greater number of cases and comparison between the different kinds of anti-migration measures must be performed.

## Figures and Tables

**Figure 1 medicina-59-02035-f001:**
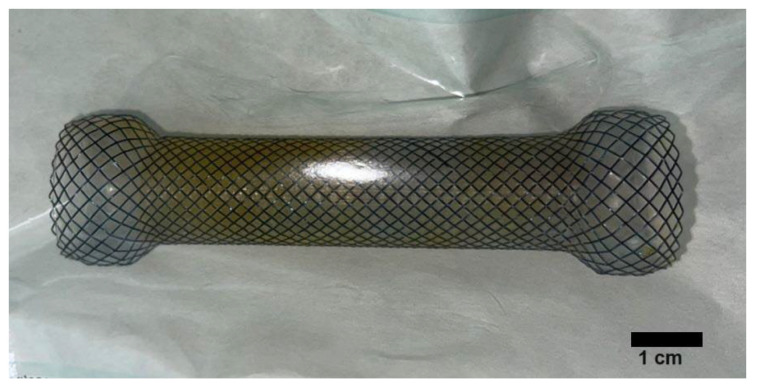
Partially covered self-expandable stent (Changzhou City Zhiye Medical Devices Institute, China).

**Figure 2 medicina-59-02035-f002:**
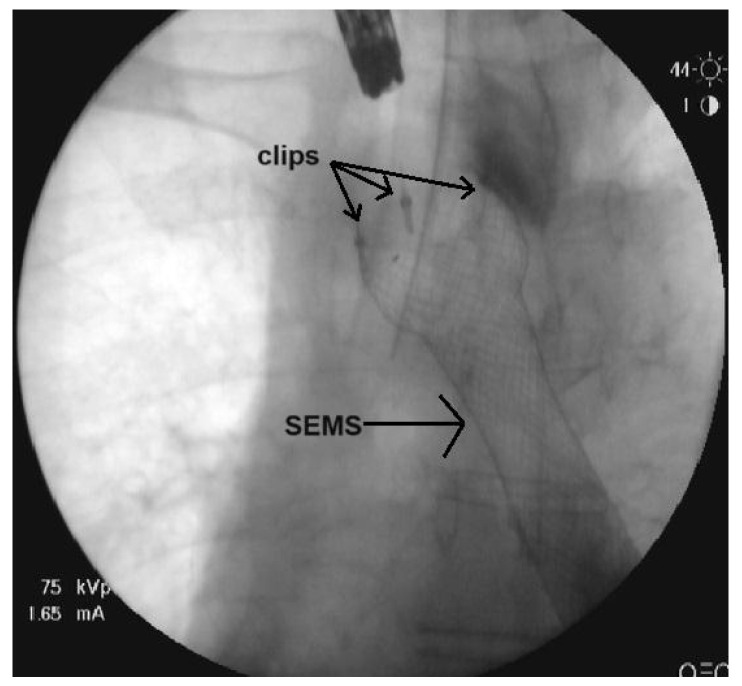
Fluoroscopy film of the SEMS fixated with clips.

**Table 1 medicina-59-02035-t001:** Patients’ data.

Case	Age	Gender	Reason for Esophageal Stenting	Location	Pretreatment
1	63	Male	Lung Cancer	Middle	Chemotherapy
2	75	Male	Adenocarcinoma	Distal	None
3	84	Male	Adenocarcinoma	Distal	None
4	78	Male	Adenocarcinoma	Distal	None
5	71	Male	Adenocarcinoma	Distal	None
6	72	Female	Lung Cancer	Middle	None
7	79	Male	Benign Stricture	Middle	None
8	74	Female	Benign Stricture	Middle	None
9	84	Male	Adenocarcinoma of the Anastomosis	Distal	Total Gastrectomy
10	58	Male	Lung Cancer	Distal	Chemotherapy
11	58	Female	Squamous cell carcinoma of the esophagus, Perforation	Middle	None
12	62	Female	Tracheoesophageal fistula post carcinoma pulmonis	Middle	Brachytherapy
13	80	Female	Adenocarcinoma	Distal	None
14	74	Male	Adenocarcinoma	Distal	None
15	46	Male	Perforation	Distal	None
16	63	Male	Adenocarcinoma	Distal	Chemotherapy
17	41	Male	Perforation	Distal	None
18	68	Female	Adenocarcinoma	Distal	None

## Data Availability

Data are contained within the article.

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
