# Peer review of "Prevention of Migration of Esophageal Self-Expandable Metallic Stents Using Endoscopic Clips"

_medicina, 2023, doi:10.3390/medicina59112035_

Round 1
Reviewer 1 Report
Comments and Suggestions for Authors
In the Abstract, Materials and Methods, and Conclusions, the authors state that stent fixation was applied to “… patients with high risk of migration of the stent”, to “… cases with higher risk for migration” and in “… cases considered high-risk for migration”.
These statements are not clear. The risk for migration should be clearly defined and the criteria for risk estimation should be reported.
Introduction
“In this study we review our experience and results in esophageal stenting and using hemostatic through the scope clips.”
Please be consisted throughout the manuscript regarding the type of clips used.
Materials and Methods
“The SEMS was fixed in its proximal end using 2 to 4 EZ through the scope clips (Olympus).”
More information on the clip type, material, number and technique of clip application should be provided.
Results section
“No other short or long-term complications were reported.”
How long was the follow-up period?
“The metallic stents were removed in a period from two weeks to two months, depending on the reason for stenting.”
Apparently stent removal applies to cases of benign conditions. Please specify and give more details..
Author Response
Cover letter reviewer 1
We kindly thank you for your comments, which will be addressed in the following cover letter.
- In the Abstract, Materials and Methods, and Conclusions, the authors state that stent fixation was applied to “… patients with high risk of migration of the stent”, to “… cases with higher risk for migration” and in “… cases considered high-risk for migration”. These statements are not clear. The risk for migration should be clearly defined and the criteria for risk estimation should be reported.
Answer: We included the conditions, that are considered with higher risk of migration in the text.
- In this study we review our experience and results in esophageal stenting and using hemostatic through the scope clips.”
Please be consisted throughout the manuscript regarding the type of clips used.
Answer: In all of the procedures we used hemostatic through the scope EZ clips developed by Olympus
- “The SEMS was fixed in its proximal end using 2 to 4 EZ through the scope clips (Olympus).” More information on the clip type, material, number and technique of clip application should be provided.
Answer: We used 2 to 4 EZ Olympus clips, depending on the type of stenosis. “The stent fixation was performed under endoscopic guidance and the way of position-ing them in the esophageal lumen, that we were aiming for was either in two opposite sides or in the resemblance of the Mercedes sign. We used two clips in nine cases, three clips – in eight cases and four clips in only one case. “
- “No other short or long-term complications were reported.” How long was the follow-up period?
“The metallic stents were removed in a period from two weeks to two months, depending on the reason for stenting.”
Apparently stent removal applies to cases of benign conditions. Please specify and give more details.
Answer: The follow up period for the two groups of patients has been provided. Moreover, in the section results the benign and the malignant cases has been presented in two different paragraphs, with different time of follow up and different complication results.
Reviewer 2 Report
Comments and Suggestions for Authors
The study presents a compelling approach to a common clinical problem, showing promise in addressing the migration of SEMS. The introduction could be enhanced by including a more extensive literature review, which would situate the research more solidly within the existing body of work. Additionally, expanding the discussion to address limitations and future research directions could provide greater depth. Language and presentation refinements are advised to elevate the overall readability and professionalism of the paper. Overall, the study is methodologically sound, offers a novel solution to an important clinical challenge, and is likely to engage the interest of the field's practitioners
Comments on the Quality of English LanguageThe manuscript is generally well-written; however, it would benefit from minor revisions to address occasional grammatical errors and improve sentence structure for enhanced clarity.
Author Response
Thank you for your review. We will take it into consideration and do better!
Reviewer 3 Report
Comments and Suggestions for Authors
Boyanov et al., address a clinically important issue related to migration of esophageal stents which is a common complication. They use endoscopic clips to prevent migration and test this procedure on patients with high risk of stent migration. While there is high relevance and immediate clinical application for using clips with esophageal stenting, there are several sections lacking details and missing information that would allow reproducibility of the work. The Materials and Methods section need to be very detailed as to how the entire procedure is performed. The Figures can be improved to include more details about the stent design with clips, images of benign and malignant cases before and after stenting. As this requires a significant amount of work, I recommend a major revision of the manuscript before consideration for publication.
Major comments:
1. Figure 1: Please include an illustration of the design of the partially covered self-expandable metallic stents showing stent fixation with endoscopic clips. The authors mention that the length of the stent was chosen in correlation with the length of stenosis. However, it is important to include details about the diameter and the range of varying lengths of the stents used in a separate section under Methods called “stent customization” for data reproducibility.
2. As this is a Brief Report, a detailed Methods section is required. What were the preoperative, stenting and postoperative procedures followed? This is a Brief Report and there are limits to the word count, but this information must be made available to the readers for reproducibility. Please include additional information in the Supplementary section.
3. What were the criteria used to determine that a patient would be at a high risk for migration?
Discuss this separately for benign and malignant cases.
3. It would be helpful to show upper gastrointestinal tract radiography images for patients before and after the stent placement to prove that stent placement using clips improves outcomes in patients with high risk of migration. Additionally, a graphical representation of the successful and no complication cases should be included for benign and malignant cases in the Figures to highlight success of the procedure.
4. In the Discussion section it would be important to discuss how using clips provides advantages over other existing and upcoming approaches in the field to prevent migration of esophageal stents.
Minor comments:
Expand ESGE in Introduction.
Comments on the Quality of English LanguageThe quality of English can be improved overall.
Author Response
We kindly thank you for your comments, which will be addressed in the following cover letter.
- Figure 1: Please include an illustration of the design of the partially covered self-expandable metallic stents showing stent fixation with endoscopic clips. The authors mention that the length of the stent was chosen in correlation with the length of stenosis. However, it is important to include details about the diameter and the range of varying lengths of the stents used in a separate section under Methods called “stent customization” for data reproducibility.
Answer: The range of lengths and the diameter of the stents were added to the text.
- As this is a Brief Report, a detailed Methods section is required. What were the preoperative, stenting and postoperative procedures followed? This is a Brief Report and there are limits to the word count, but this information must be made available to the readers for reproducibility. Please include additional information in the Supplementary section.
Answer: The stenting and clip fixation were thoroughly explained and the postprocedural and follow up procedures were added.
- What were the criteria used to determine that a patient would be at a high risk for migration?
Discuss this separately for benign and malignant cases.
Answer: The conditions considered at a higher risk for migration were added to the introduction. We have used available literature data and have not developed our own criteria.
- It would be helpful to show upper gastrointestinal tract radiography images for patients before and after the stent placement to prove that stent placement using clips improves outcomes in patients with high risk of migration. Additionally, a graphical representation of the successful and no complication cases should be included for benign and malignant cases in the Figures to highlight success of the procedure.
Answer: We do not usually perform radiography images before the stenting, only after.
- In the Discussion section it would be important to discuss how using clips provides advantages over other existing and upcoming approaches in the field to prevent migration of esophageal stents.
Answer: Since we have a small retrospective study, that only includes one option for reducing the risk of migration of esophageal SEMS. We did not feel comfortable to compare it to other techniques.
Round 2
Reviewer 3 Report
Comments and Suggestions for Authors
While the authors have addressed the comments adequately, some details must be added to improve quality.
1. Please add scale bar in Figure 1.
2. Please label Figure 2 - oesophagus, stent, clips.
3. Please correct the grammatical error in the sentence: "In this report we aim to review our experience and results in stent fixation with through the score clips.."
4. Please pay attention to a/an/the
According to "the" European Society of Gastrointestinal Endoscopy’s (ESGE) latest guideline.
5. "In our study, only one of eighteen procedures resulted in stent migration, that shows even better results than the above-mentioned study"
Cite reference for "the above mentioned study". What is the percentage success rate?
6. "Typically, fully covered SEMSs are more prone to migration than 104 the partially covered and the non-covered ones". Please discuss why fully covered SEMs are more prone to migration or cite an appropriate reference.
Comments on the Quality of English LanguageSpecial attention needs to be paid to sentence construction throughout the manuscript. There are too many errors and the text doesn't read well most of the places.
Author Response
Thank you for your comments. We have taken all of them into consideration and would like you to review the revised version of the manuscript